# Exploring Selective Layer Freezing Strategies in Transformer Fine-Tuning: NLI Classifiers with Sub-3B Parameter Models

## Abstract

In recent years, methods that selectively fine-tune or reduce the number of layers in large language models (LLMs) have garnered attention as an efficient alternative to traditional fine-tuning, where all layers are trained. In this paper, we revisit the concept of **Layer Freezing**, a simple yet effective fine-tuning strategy, and introduce detailed strategies that improve the training efficiency of LLMs by selectively fine-tuning only a portion of the layers. We tested various freezing ratios and positions, and found that by freezing the bottom 25% or 50% of transformer layers during fine-tuning of an LLM with sub 3 billion parameters, we can achieve performance equal to or better than full model fine-tuning and Low-Rank Adaptation (LoRA), while significantly reducing memory usage and training time. Our experiments on natural language inference tasks show that this approach reduces memory consumption by about 30% and 50%, and improves training speed by 20-30%.

## 1 Introduction

Recent advancements in Natural Language Processing (NLP) have been largely driven by the emergence of Large Language Models (LLMs) such as GPT-3 (Brown et al., 2020), PaLM (Chowdhery et al., 2023), and LLaMA (Touvron et al., 2023). These models leverage extensive pre-training to learn a wide range of linguistic patterns and knowledge. They demonstrate high performance across various tasks through techniques like prompt engineering (Liu et al., 2023) and in-context learning (Xie et al.). As a result, LLMs have become indispensable tools in numerous NLP tasks, including translation, question answering, and document generation.

However, several limitations are associated with the training and application of LLMs: First, high-performing LLMs typically contain over 7 billion parameters, making them computationally expensive and requiring vast resources and time for both training and inference. Additionally, LLMs tend to exhibit inconsistent performance in unfamiliar domains or tasks that were not encountered during pre-training (Hendrycks et al., 2020). This is a chronic problem for pre-trained models, which are constrained to generating outputs based on their pre-trained knowledge. To address this fundamental issue, it is imperative to implement knowledge updates through fine-tuning processes, enhancing the model's adaptability (Gururangan et al., 2020). However, due to the immense size of modern LLMs, with parameter counts ranging from billions to hundreds of billions, even fine-tuning demands significant computational effort.

To address these computational challenges, we revisit the concept of **Layer Freezing**, a simple yet effective fine-tuning strategy, and introduce detailed strategies that extend this approach. Previous studies have explored freezing layers of small language models such as BERT (Devlin et al., 2019) during fine-tuning, but these efforts have mainly focused on improving speed and have encountered challenges due to the complexity of freezing techniques (Ben Zaken et al., 2022; Tang et al., 2024).

In contrast, we have found that simply freezing a subset of layers can achieve better computational efficiency and superior performance compared to fine-tuning the entire model. Rather than introducing additional layers or parameters, our goal is to reduce costs and maximize training efficiency by focusing on fine-tuning only a subset of layers within the existing LLM. The discovered method has the following advantages:

- **Simplicity**: This approach is highly straightforward and can be easily applied without the need for complex analysis or modifications to the model architecture.

- **Universality**: This method can be widely applied across various model architectures, regardless of scale or structural complexity.

- **Performance Improvement**: Experimental results show that this method not only improves computational efficiency but also enhances model performance compared to fine-tuning all layers.

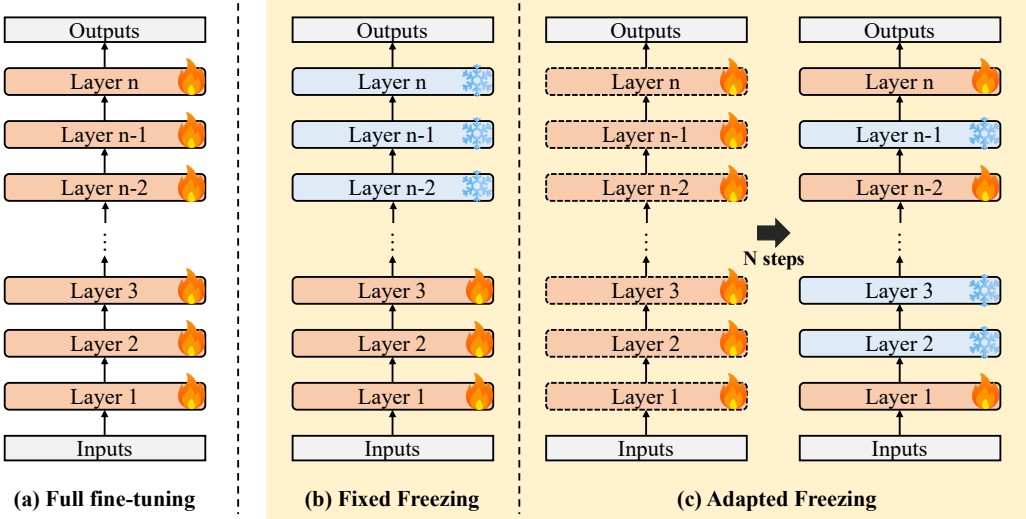

Figure 1: Proposed freezing strategies. (a) is the conventional fine-tuning method, while (b) and (c) are the proposed strategies. In (b), predetermined layers are frozen according to preset configurations before training. In (c), after initial training, layers to freeze are selected based on the amount of weight change.

Figure 1 illustrates a comparison between the conventional fine-tuning method that utilizes all layers and the layer selection method tested in this study. In the Fixed Freeze scenario, a predetermined ratio and location of layers are frozen before training begins. In contrast, the Adapted Freeze approach involves recording and analyzing the initial training process, selecting the appropriate layers to freeze, and then completing the remaining training.

To validate our approach, we focused on LLMs with fewer than 3 billion parameters, which can be trained on a single GPU. We conducted a comparative analysis using the Natural Language Inference (NLI) task (Bowman et al., 2015), a subset of text classification problems. The NLI task was chosen as it effectively assesses a model's fundamental language understanding and reasoning capabilities.

In our research, we discovered a remarkably effective method to address these challenges. By freezing the bottom 25% of layers in transformer models during fine-tuning, we achieved significant improvements in both computational efficiency and model performance. This approach reduced training memory usage by over 30% (excluding model memory) compared to full model fine-tuning, while generally enhancing overall performance. Additionally, we observed a notable 20% increase in training speed. Empirical comparisons revealed that this method demonstrated superior performance metrics relative to Low-Rank Adaptation (LoRA) (Hu et al., 2021), while exhibiting comparable levels of computational acceleration and memory reduction.

## 2 RELATED WORK

### 2.1 NATURAL LANGUAGE INFERENCE

NLI, also known as recognizing textual entailment (Dagan et al., 2005), is a fundamental task in NLP. This task involves determining the logical relationship between a premise and a hypothesis.

Specifically, a model must classify this relationship as entailment (the hypothesis necessarily follows from the premise), contradiction (the hypothesis contradicts the premise), or neutral (the hypothesis may or may not be true given the premise).

NLI serves as a crucial indicator for evaluating a model's language understanding and reasoning capabilities, including comprehension of semantics, context, and logical relationships. Models that demonstrate high performance in NLI tasks tend to excel in other language understanding tasks as well. Success in NLI challenges often indicates a level of language understanding that generalizes across various domains and linguistic tasks (Poliak et al., 2018). Consequently, NLI has established itself as a valuable benchmark for assessing language models (Wang et al., 2018).

## 2.2 GENERAL FINE-TUNING APPROACHES

The traditional fine-tuning approach involves retraining all or some of the parameters of a pre-trained model to adapt it to a new task (Howard & Ruder, 2018). This method adjusts the weights of the pre-trained model, utilizing various strategies such as learning rate adjustment, gradual unfreezing, and discriminative learning rates (Peters et al., 2019). While this is useful for training task-specific models, it has limitations in terms of computational cost and resource efficiency when fine-tuning all parameters in recent large-scale models. To address these issues, techniques that reduce the number of trainable parameters, such as LoRA, or reduce memory usage, such as LLM quantization (Dettmers et al., 2022), are gaining attention.

## 2.3 LAYER FREEZING IN FINE-TUNING

Since the emergence of language models, there have been attempts to improve the efficiency of fine-tuning by freezing certain layers. Ben Zaken et al. (2022) proposed a method that fine-tunes only the bias parameters instead of the weights in transformer-based masked language models, reducing memory usage and improving speed. Tang et al. (2024) introduced a technique to accelerate the training process by gradually freezing layers based on their impact during training. However, both studies primarily focus on speed improvement, which often leads to performance degradation. Additionally, these studies require complex mechanisms for layer freezing.

In contrast, our study employs a simple freezing method that can be applied to any model while also demonstrating performance improvement. This straightforward approach stands out in that it not only reduces computational cost but also enhances model performance.

## 3 LAYER SELECTION FOR FINE-TUNING

This study proposes a method to improve model training efficiency by fine-tuning only a subset of layers in an LLM and aims to validate it experimentally. In contrast to conventional LLM fine-tuning methods that involve training all layers, this study demonstrates that selectively freezing specific layers and fine-tuning only the remaining ones can lead to improved performance, faster learning speed, and reduced memory usage.

### 3.1 FIXED FREEZING

The main strategy used during the model training process is to freeze specific layers of the model and fine-tune only the remaining layers. We evaluated the impact of various layer selection methods on model performance and training efficiency.

- **Bottom-Up Freezing**: We experimented with a method that sequentially freezes the model's bottom layers, allowing only the remaining upper layers to be fine-tuned. As the bottom layers are primarily responsible for the basic linguistic expressiveness of the language model, while the upper layers tend to learn task-specific representations (Rogers et al., 2020), we hypothesized that this freezing approach would preserve fixed linguistic knowledge while enabling task-specific adaptation.
- **Top-Down Freezing**: We tested an approach that freezes the top layers and trains only the bottom layers. This method anticipates that the bottom layers will be tuned to the task based on the fixed higher-level concepts in the frozen upper layers.

- **Interval Freezing**: This method involves freezing layers at intervals of $n$, meaning that every $n$-th layer is frozen during training. This approach aims to allow both upper and lower layers to be appropriately adjusted simultaneously, encouraging information to be learned evenly across various layer levels.

## 3.2 ADAPTED FREEZING

As an alternative to the Fixed Freezing strategy, we propose an Adaptive Freezing approach with dynamic layer selection. In this approach, we track the weight changes of each layer during training to identify layers with significant or minimal changes. Based on these changes, we automatically identify the layers that play a crucial role in performance. The Top-$N$ layers, according to the magnitude of weight changes, are then selectively frozen before proceeding with training. The following outlines the operational sequence of this adaptive layer selection method:

1. **Weight Change Tracking**: We calculate the change in weights for each layer by comparing the layer-wise weights before training and after the first 5 steps of training. The magnitude of weight changes for multiple parameters within a single layer was quantified as a single scalar value by computing the L2 norm of the changes and then taking the mean across all parameters in the layer.

2. **Top-$N$ Layer Selection**: We select the top $N$ layers with either the largest or smallest weight changes, freeze them, and then resume training. Through this, we aimed to understand the roles that layers with large and small weight changes play in the fine-tuning.

## 3.3 FREEZING STRATEGIES

Through these Fixed and Adapted Freezing methods, we aim to experimentally demonstrate that fine-tuning only a subset of layers can reduce memory and computational costs compared to fine-tuning the entire model. We hypothesize that this approach can potentially improve performance or, at minimum, maintain it without degradation.

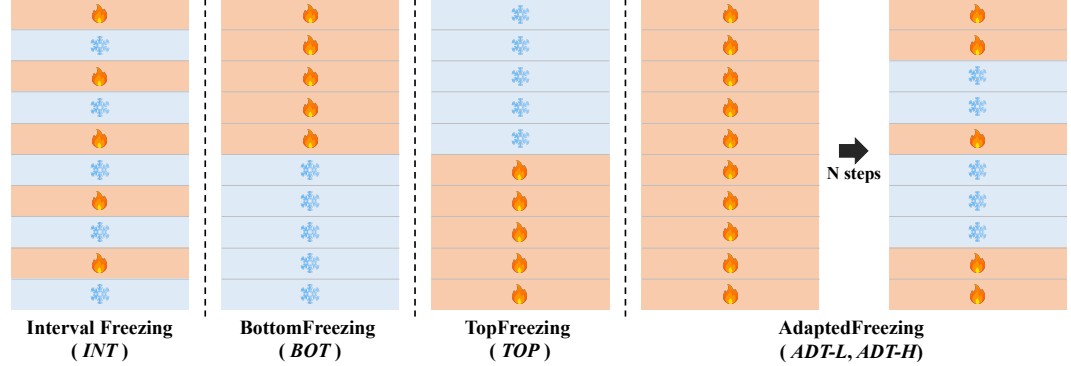

Figure 2: Proposed detailed freezing strategies, with 50% freezing for each strategy.

The following are abbreviations for the freezing strategies used in this study:

- $ALL$: Fine-tuning using all layers, used as the baseline.
- $LoRA$: Fine-tuning using LoRA, also used as the baseline.
- $INT$ (**Interval**): Fine-tuning by freezing layers at regular intervals.
- $BOT$ (**Bottom-Up**): Fine-tuning by freezing layers starting from the bottom of the model.
- $TOP$ (**Top-Down**): Fine-tuning by freezing layers starting from the top of the model.
- $ADT$-$L$ (**Adapted Low**): Fine-tuning by freezing $N$ layers with the smallest weight changes.
- $ADT$-$H$ (**Adapted High**): Fine-tuning by freezing $N$ layers with the largest weight changes.

The number following each abbreviation indicates the percentage of frozen layers. For example, $INT25$ means 25% of the layers are frozen at regular intervals, while $TOP50$ means 50% of the layers are frozen starting from the top.

Figure 2 visualizes the freezing strategies utilized in this study. When 50% of the total layers are frozen, the layers are frozen and trained in the pattern shown in the figure for each strategy.

## 4 EXPERIMENTS

The primary objective of this study is to verify whether fine-tuning only a subset of layers in LLMs can achieve sufficient training effectiveness compared to training all layers. Through our approach, we aim to explore a methodology that reduces memory and computational resource requirements while maintaining training effectiveness without performance degradation.

### 4.1 MODELS AND DATASETS

The experiments used decoder-only small LLMs with sub 3 billion parameters, such as Gemma-2b(Gemma) (Team et al., 2024), Phi-2 (Javaheripi et al., 2023), and MiniCPM-2b-128k(MiniCPM) (Hu et al., 2024), which can be trained on a single GPU. All these models are large-scale pre-trained language models whose parameters can be fine-tuned for specific tasks.

For the experiments, we used NLI tasks, which are primarily text classification problems designed to verify the models' basic language understanding and reasoning abilities by determining logical relationships between sentences. NLI-related tasks from the GLUE (Wang et al., 2018) and Super-GLUEWang et al. (2019) benchmarks were used. The specific tasks are listed in Table 1.

| Dataset | Task | Number of samples | Usage rate |
|---------|------|-------------------|------------|
| GLUE | RTE | 2,500 | 100% |
| GLUE | QNLI | 3,200 (105,000) | 3.0% |
| GLUE | WNLI | 635 | 100% |
| GLUE | MNLI | 3,200 (392,000) | 0.8% |
| SuperGLUE | CB | 250 | 100% |

Table 1: Number of samples in each dataset, the number in parentheses is the total number of data, but we used only a maximum of 3,200 randomly selected data.

### 4.2 EXPERIMENTAL DESIGN

In this study, we conducted a series of experiments to evaluate the efficacy of fine-tuning strategies that selectively utilize specific layers of neural networks. Our experimental design focused on various freezing techniques, enabling a comparative analysis of performance variations resulting from each approach. The primary objective of these experiments was to conduct a comprehensive assessment of model performance, memory utilization, and training efficiency to determine the optimal freezing methodology.

**Finding the Optimal Freezing Ratio:** First, we conducted experiments to determine at which ratio of frozen layers the model exhibits the highest performance. To achieve this, we froze a certain proportion of the model's layers and trained only the remaining layers. For the $INT$ strategy, we applied freezing ratios of 25%, 33.3%, and 50%, while for the $BOT$, $TOP$, $ADT$-$L$, and $ADT$-$H$ strategies, we used ratios of 25%, 50%, and 75%. Through this approach, we aimed to identify the threshold at which performance sharply declines when more than a certain proportion of layers are frozen, thereby deriving the optimal freezing ratio.

**Finding the Optimal Freezing Position**: Next, we analyzed which positions within the model's layers have the most significant impact on performance when frozen. The freezing positions were categorized as $BOT$ (Bottom-Up; freezing layers sequentially starting from the lower layers), $TOP$ (Top-Down; freezing layers sequentially starting from the upper layers), and $INT$ (Interval; freezing layers at regular intervals). By comparing the performance for each freezing position, we examined

which layers play more crucial roles. Additionally, we tracked the weight changes during the training process and adaptively froze layers to understand the significance of the weight changes.

**Finding the Optimal Freezing Strategy:** Finally, we aimed to identify the optimal freezing strategy by comprehensively considering the performance, training speed, and memory usage of each strategy. As it is challenging to fairly and objectively quantify these diverse aspects into a single metric, we first evaluated each aspect quantitatively and then conducted a comprehensive analysis.

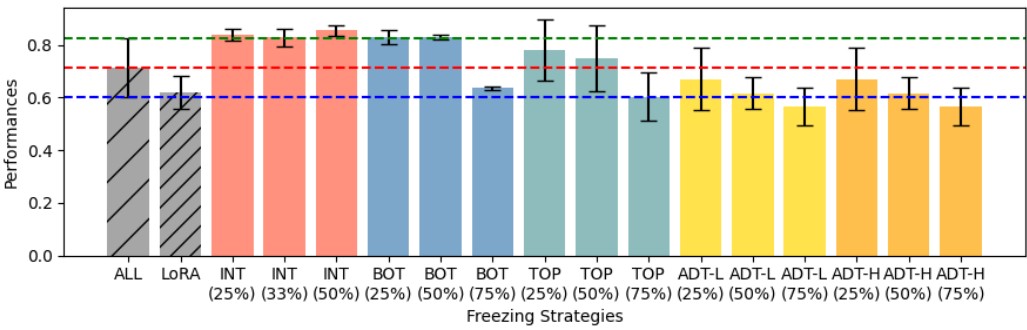

(a) Accuracy performance of Gemma on the QNLI task

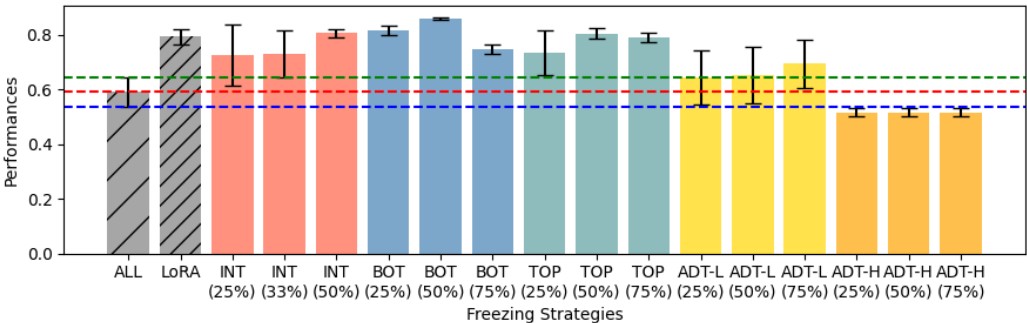

(b) Accuracy performance of Phi-2 on the QNLI task

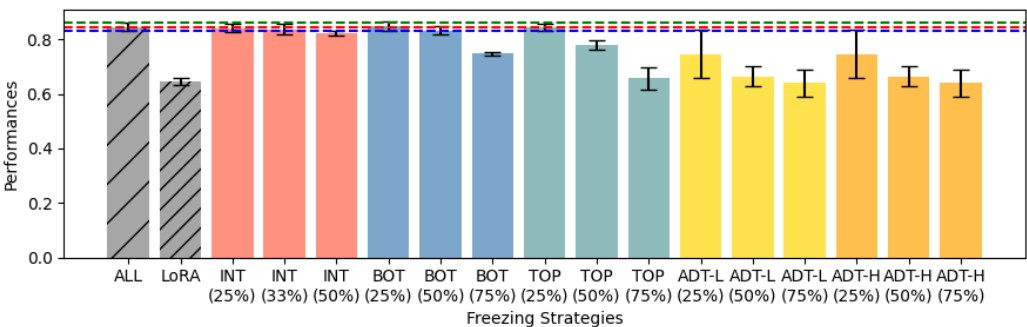

(c) Accuracy performance of MiniCPM on the QNLI task

Figure 3: Accuracy performance on the QNLI task. The same colors indicate the same strategy. The gray hatched bar represents the baseline performance achieved by fine-tuning all layers of the model. The red dashed line represents the average performance of fine-tuning across all layers, The blue dashed line represents the mean performance minus one standard deviation, while the green dashed line shows the mean performance plus one standard deviation when all layers are fine-tuned.

### 4.3 HYPER-PARAMETER SETTING

For each strategy, we conducted training using five different random seeds: 42, 43, 44, 45, and 46, and measured the average performance. We fixed the batch size at 32 and max length at 128, and trained the model for a total of 100 steps for each experiment. The learning rate is $5e^{-5}$ and we used cosine learning rate scheduler. For the LoRA implementation in our experiments, we set the rank ($r$) to 8 and the alpha parameter to 32. A dropout rate of 0.1 was applied, and no quantization was performed. All other hyper-parameters remained consistent across experiments.

### 4.4 PERFORMANCE COMPARISON BY FREEZING STRATEGY

Figure 3 shows the average performance of the three models on the QNLI task, measured five times using different random seeds for various freezing strategies. The error bars represent the $\mu \pm \sigma$ (mean $\pm$ standard deviation). The gray bars represent the experimental results for the baseline approaches: full fine-tuning $ALL$ and $LoRA$. The red dashed line indicates the mean performance when all layers are fine-tuned. The blue dashed line represents the mean performance minus one standard deviation, while the green dashed line shows the mean performance plus one standard deviation when all layers are fine-tuned.

In most cases, the $INT25$, $BOT25$, and $TOP25$ strategies showed superior performance compared to the $ALL$ and $LoRA$ strategy. Notably, the Phi-2 model achieved even higher performance, especially with the freezing strategies. The $BOT$ strategy consistently demonstrated excellent and solid performance across all models and most tasks. Furthermore, the $INT$ strategy generally showed lower standard deviation compared to $ALL$, indicating more stable learning. Contrary to expectations, the Adapted Freezing strategy did not show performance merits. Furthermore, freezing layers with either high or low weight changes did not yield significant results. Total experimental results can be found in Appendix A.

### 4.5 TRAINING EFFICIENCY COMPARISON BY FREEZING STRATEGY

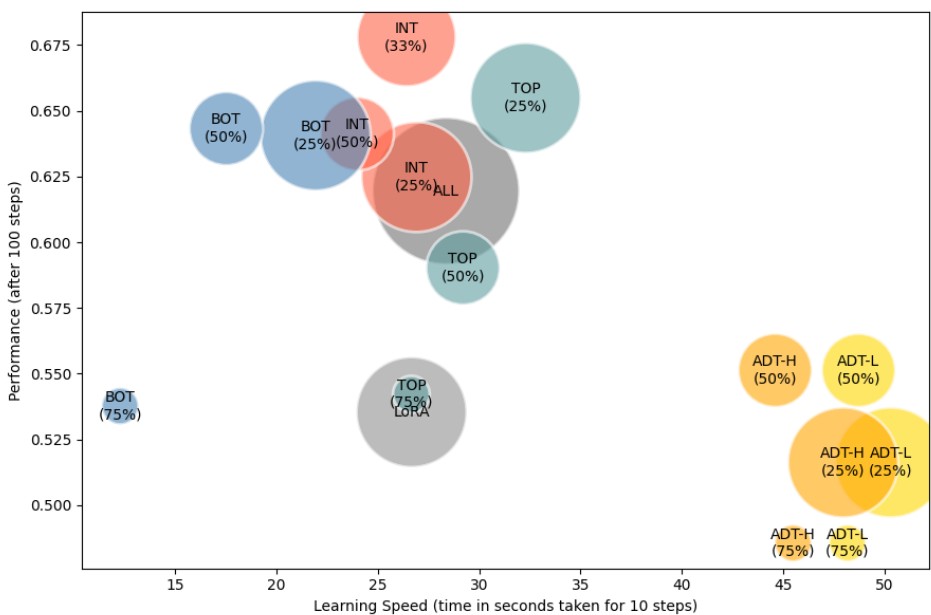

Figure 4: Learning speed and CB performance of the Gemma model for different freezing strategies. The size of the semicircles represents the relative proportion of unfrozen layers used in training. The x-axis indicates the time (in seconds) taken for 10 steps of training, and the y-axis represents the performance after 100 steps of training. The $TOP$ strategy is depicted with green circles, $BOT$ with blue, $INT$ with red, $ADT$-$L$ with yellow, and $ADT$-$H$ with orange circles.

Figure 4 shows the average performance and training time measured by training the Gemma model on the MNLI task for 10 steps using five different random seeds. The size of the circles visually represents the proportion of layers that were frozen. From this, it can be observed that the $INT$, $BOT$, and $TOP$ strategies learn faster than the $ALL$ strategy. In particular, the $BOT$ strategy generally showed very fast learning speed, comparable to the $TOP$ and $INT$ strategies. Moreover, when compared to $LoRA$, the $BOT$ strategies showed comparable or superior learning speed. $ADT$-$L$ and $ADT$-$H$, on the other hand, exhibited relatively slower speeds due to an additional fixed time of about 13 seconds required for selecting the initial layers to freeze. With sufficiently long training times, this overhead becomes negligible, and their speed is expected to be comparable to that of the $INT$ and $TOP$ strategies. Additional experimental results can be found in Appendix A.

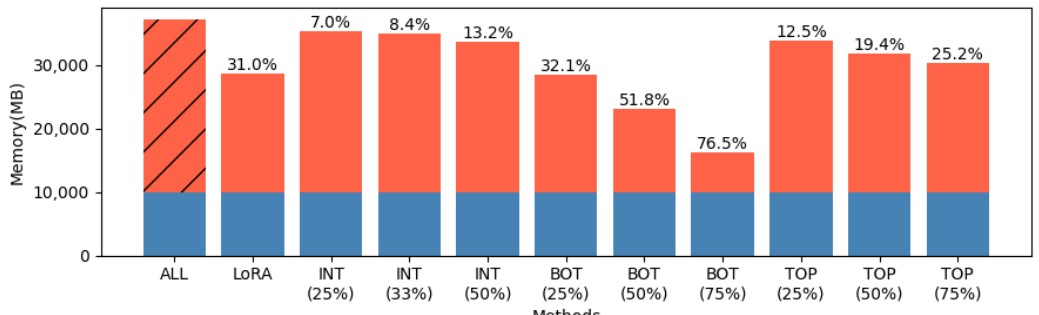

Figure 5: GPU training memory usage for each freezing strategy. The blue color represents the memory statically used by the model, while the red color indicates the memory utilized during the training process. The numbers above each bar indicate the percentage reduction in GPU memory usage compared to the $ALL$ strategy.

Figure 5 shows the GPU memory usage of the $INT$ and $BOT$ strategies compared to $ALL$. Excluding the memory used by the fixed model, when comparing only the training memory, the $INT$ strategy used approximately 7%–13% less memory and the $BOT$ strategy used approximately 12%–25% less. In the case of $BOT$, as the freeze ratio increased, the GPU memory usage dramatically decreased by about 20% at each ratio. Additionally, $BOT25$ demonstrated a level of memory reduction comparable to that of $LoRA$. The smaller reduction in $INT$ is presumed to be due to inefficient computation caused by lack of optimization during CUDA operations, depending on the location of the frozen layers.

### 4.6 Best Freezing Strategy

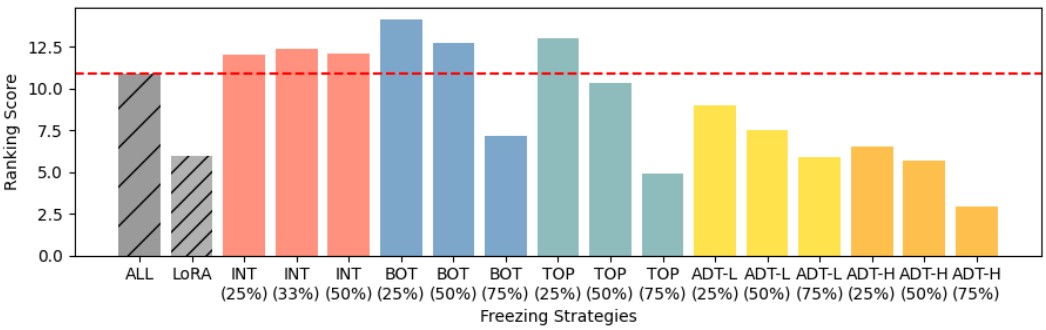

Figure 6: Overall evaluation based Ranking Score by strategy. An overall evaluation was conducted by measuring the average ranking scores based on the reversed rankings assigned across all models, datasets, and experimental settings used in the experiments. The red dashed line represents the score of the $ALL$ strategy.

**Ranking Score = Mean(reversed rank):** To determine the optimal strategy, we assigned scores based on the performance rankings from our experiments. In this paper, a total of 17 experiments were conducted using $ALL$ and $LoRA$ strategy and 15 freezing strategies. Accordingly, for each task, the strategy that achieved the highest average performance was given 16 points, while the strategy with the lowest average performance received 1 point. This scoring was performed for each of the 15 experiments (3 models $\times$ 5 tasks), and the final average score was calculated.

Figure 6 shows the average scores measured in this manner, and it can be observed that $BOT25$ and $TOP25$ achieved the highest scores. Notably, the $INT$ strategy outperformed the $ALL$ strategy across all ratios. This confirms that strategies $INT25$, $INT33$, $INT50$, $BOT25$, $BOT50$, and $TOP25$ can be used as alternatives to the $ALL$ strategy. Considering memory usage and training speed, the $BOT25$ and $BOT50$ strategies are judged to be the most effective.

## 5 DISCUSSION

Our study on NLI tasks has uncovered a remarkably simple yet effective approach to enhance learning efficiency in LLMs. Through our experiments, we have discovered that fine-tuning only specific layers of a 3B parameter LLM can yield performance that matches or even surpasses that of full model fine-tuning.

**Enhanced Performance:** Specifically, we found that freezing the bottom 25% or 50% of transformer layers during fine-tuning not only maintained high performance but often exceeded the results of full model fine-tuning and LoRA. This approach led to a substantial reduction in memory usage, approximately 30% and 50% respectively, without compromising model effectiveness. Notably, the training speed increased by 20-30%, which can be attributed to the reduced computational load. we posit that this phenomenon may be attributed to the model's capacity being disproportionately large relative to the complexity of the NLI task. This aligns with observations in techniques like LoRA, where freezing the majority of the model and training only a small number of additional parameters can lead to performance improvements.

**Memory Reduction:** The reduction in memory usage observed with our partial fine-tuning approach is logically consistent with the decreased number of trainable parameters. However, we noticed that interval freezing strategies, where layers are frozen in a distributed pattern throughout the model, did not yield significant memory savings. This suggests that contiguous freezing of layers is more beneficial for memory optimization. Additionally, freezing the top layers of the model resulted in less pronounced memory savings compared to freezing bottom layers. We conjecture that this discrepancy may be related to CUDA optimization techniques and underlying hardware architectures.

**Learning Speed Improvements:** While the speed improvements did not completely match the reduction in memory usage, the observed 20-30% increase in training speed is nonetheless significant. We attribute this to the substantial computational overhead inherent in processing large language models.

## 6 CONCLUSION

In this study, we investigated the effectiveness of fine-tuning only a portion of the layers in large language models (LLMs) for natural language inference (NLI) tasks. Our experiments, conducted on an LLM with approximately 3 billion parameters, demonstrated that freezing the bottom 25% or 50% of transformer layers can achieve performance equal to or better than full model fine-tuning and LoRA, while significantly reducing memory usage and increasing training speed. This indicates that our simple application of layer freezing, despite being an existing methodology, is particularly effective for NLI tasks. Our approach offers a practical and efficient strategy for utilizing large LLMs in resource-constrained environments.

Future work will focus on extending this method to a broader range of NLP tasks, including text classification, named entity recognition, and machine translation, to assess its generalizability. We aim to investigate the scalability of this approach by applying it to larger models with parameters in the hundreds of billions. Furthermore, we intend to explore the synergistic effects of combining this method with other efficiency-enhancing techniques such as quantization and pruning.

# 7 LIMITATION

This study demonstrated the efficiency of fine-tuning only certain layers in LLMs. However, the following limitations exist:

**Experiments Limited to Small LLMs:** This study primarily conducted experiments on small-scale LLMs with 3 billion parameters or fewer, such as Gemma, Phi-2 and MiniCPM. This choice was due to the experimental conditions set to enable training in a single GPU environment. Therefore, further research is needed to determine whether the proposed methodology demonstrates similar performance improvements and efficiency in extremely large models (such as PaLM and LLaMA). In extremely large models, memory requirements or training patterns may differ, necessitating experiments on these models to expand the scope of this research.

**Dataset Limited to NLI Tasks:** This study focused on Natural Language Inference (NLI) tasks, conducting experiments only on NLI-related datasets (such as RTE and CB) from GLUE and Super-GLUE benchmarks. While NLI tasks are specialized in evaluating a model's ability to infer logical relationships, other types of tasks (e.g., text generation, question answering, translation, etc.) may have different model characteristics and learning requirements. Therefore, further experiments on diverse tasks and datasets are necessary to assess the effectiveness of the proposed fine-tuning method across a broader range of natural language processing tasks.

To address these limitations, future studies should validate the generalizability and efficiency of the proposed methodology across a range of larger-scale LLMs and diverse tasks.

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

# A  MODEL PERFORMANCE

This section presents an analysis of model performance, focusing on the effects of freezing methods and dataset selection. Figures 7 and 8 show the performance and learning speed of the Gemma model on the RTE, CB, QNLI, and WNLI datasets. Figure 9 shows Gemma model performance on RTE, CB, WNLI, and MNLI dataset. Figure 10 and 11 shows the each Phi-2 and MiniCPM model, respectively, on the same datasets.

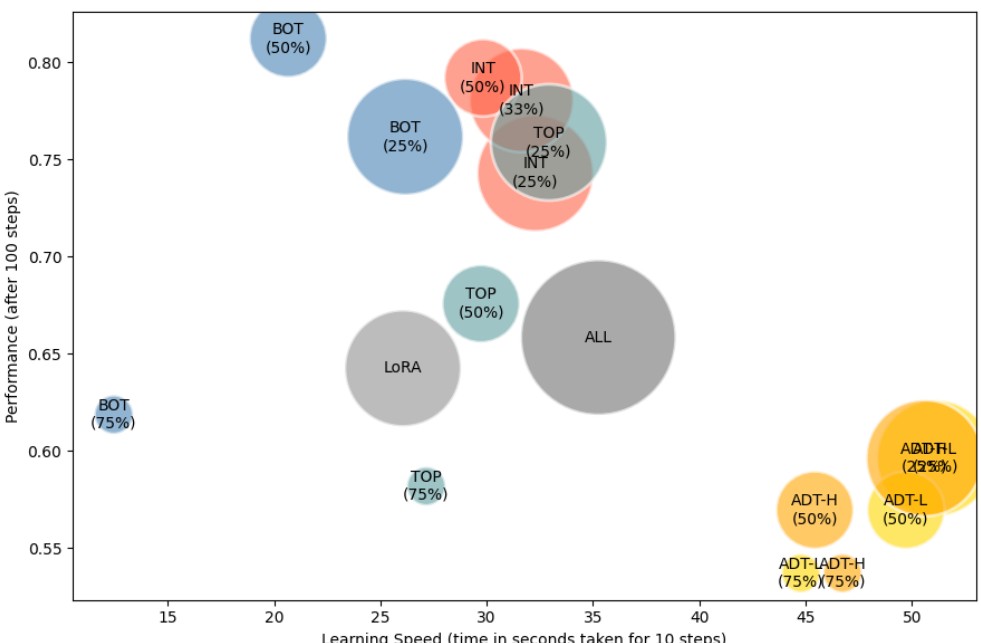

(a) Learning speed and performance of Gemma on the RTE task

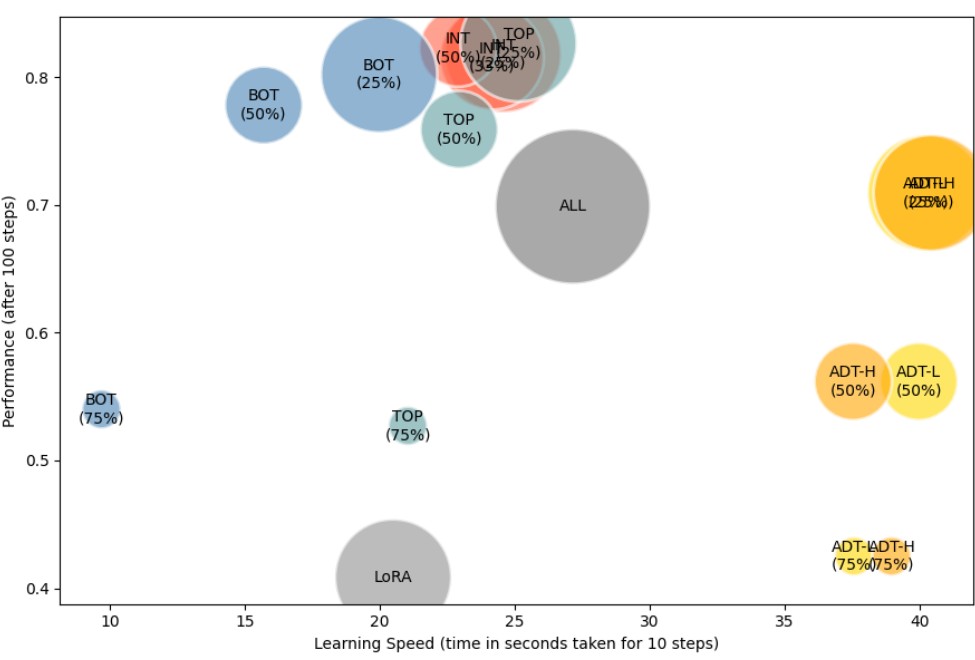

(b) Learning speed and performance of Gemma on the MNIL task

Figure 7: Learning speed and performance of Gemma on the RTE and MNIL tasks

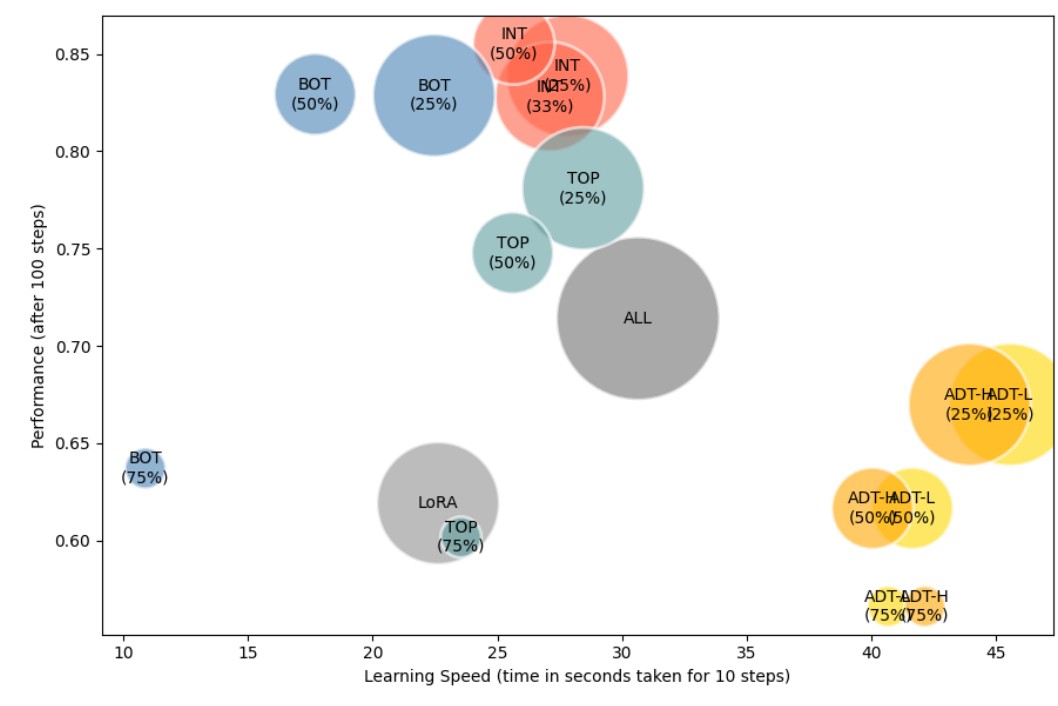

(a) Learning speed and performance of Gemma on the QNLI task

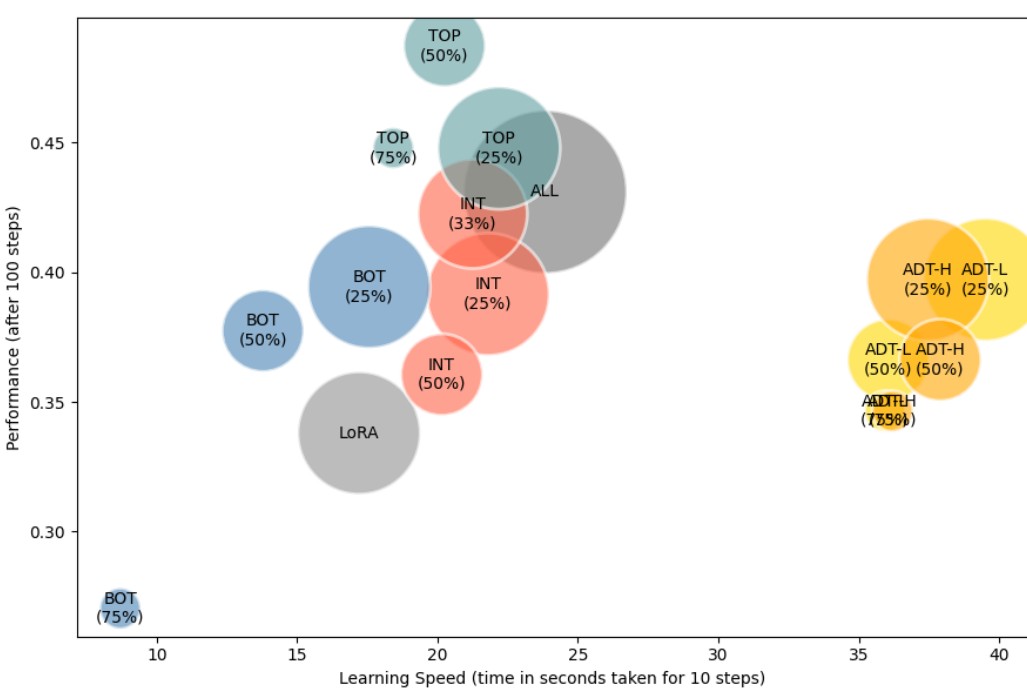

(b) Learning speed and performance of Gemma on the WNLI task

Figure 8: Learning speed and performance of Gemma on the QNLI and WNLI tasks

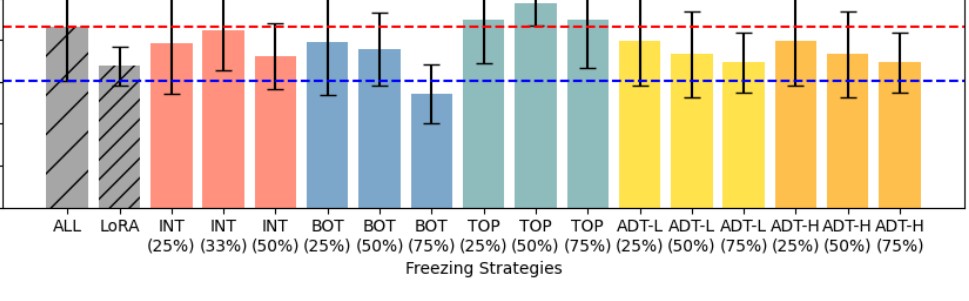

(a) Accuracy performance of Gemma on the RTE task

(b) F1-score performance of Gemma on the CB task

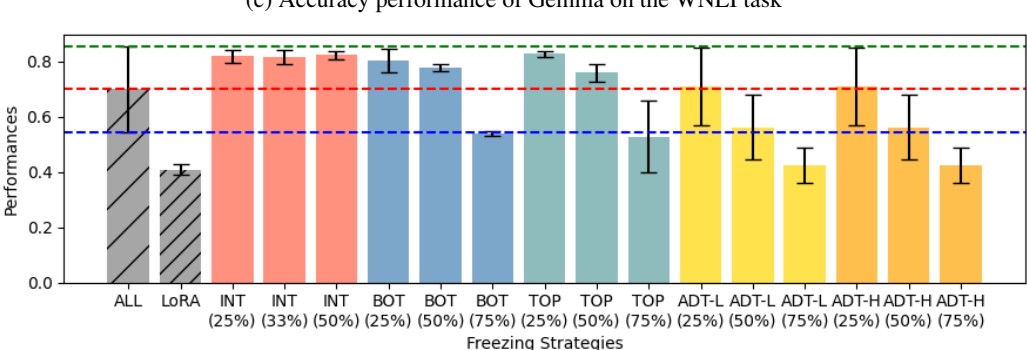

(c) Accuracy performance of Gemma on the WNLI task

(d) Accuracy performance of Gemma on the MNLI task

Figure 9: Accuracy performance of Gemma. The same colors indicate the same strategy. The gray hatched bar represents the baseline performance achieved by fine-tuning all layers of the model. The red dashed line represents the average performance of fine-tuning across all layers, The blue dashed line represents the mean performance minus one standard deviation, while the green dashed line shows the mean performance plus one standard deviation when all layers are fine-tuned.

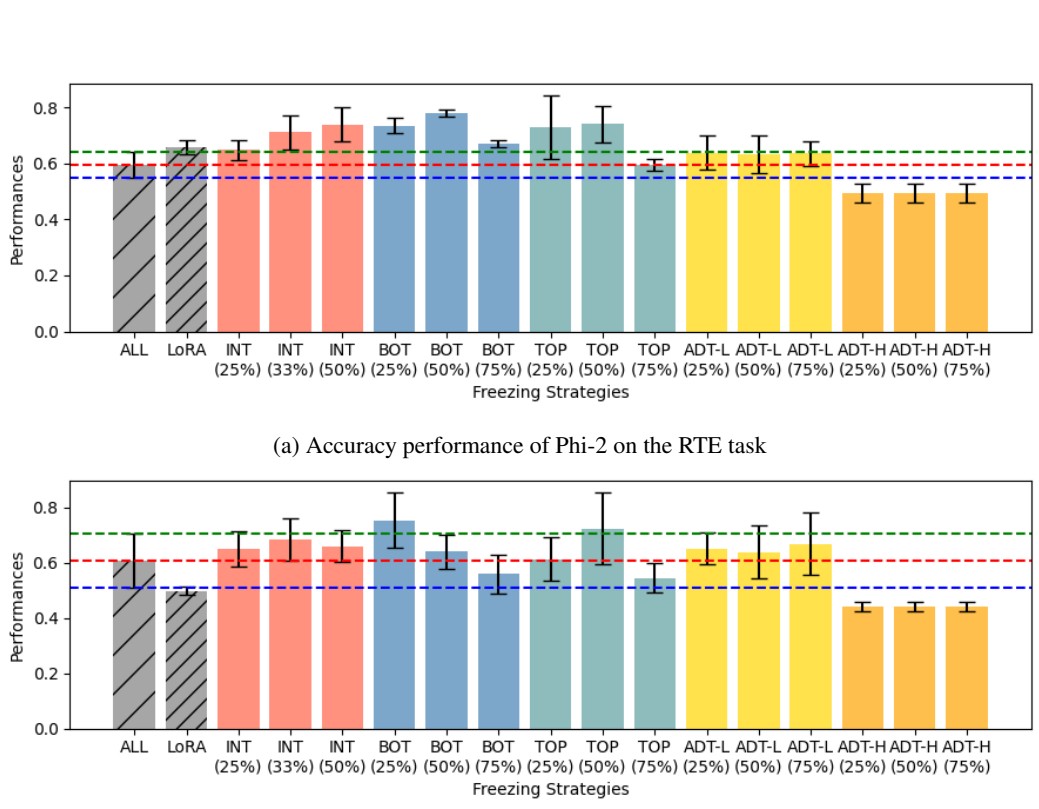

(a) Accuracy performance of Phi-2 on the RTE task

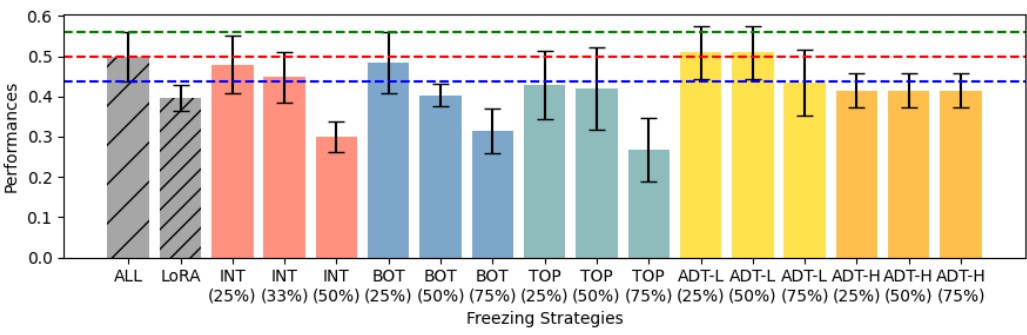

(b) F1-score performance of Phi-2 on the CB task

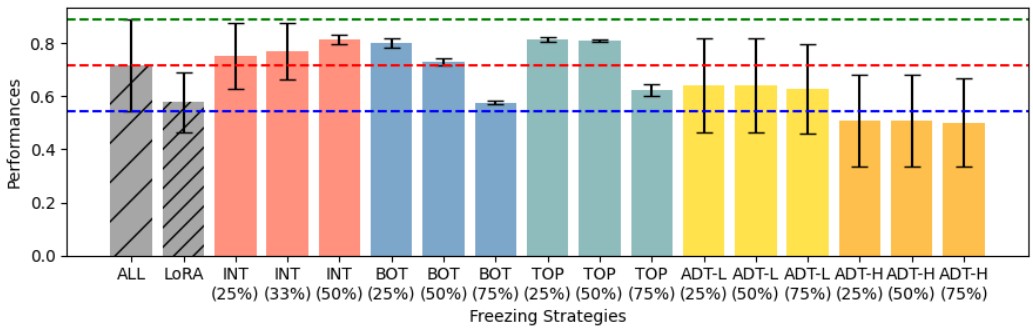

(c) Accuracy performance of Phi-2 on the WNLI task

(d) Accuracy performance of Phi-2 on the MNLI task

Figure 10: Accuracy performance of Phi. The same colors indicate the same strategy. The gray hatched bar represents the baseline performance achieved by fine-tuning all layers of the model. The red dashed line represents the average performance of fine-tuning across all layers, The blue dashed line represents the mean performance minus one standard deviation, while the green dashed line shows the mean performance plus one standard deviation when all layers are fine-tuned.

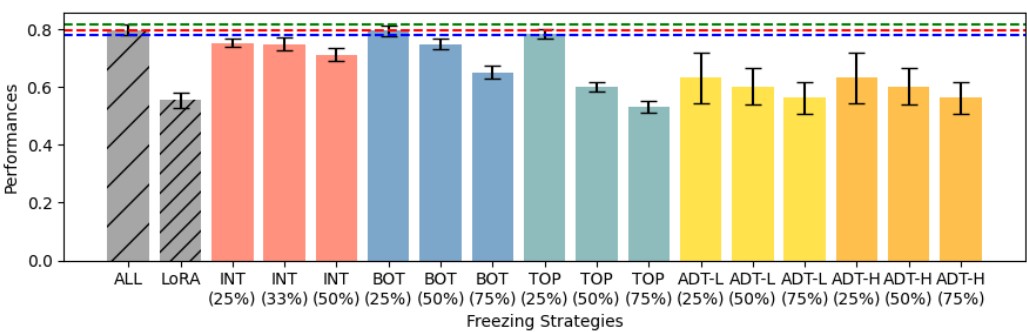

(a) Accuracy performance of MiniCPM on the RTE task

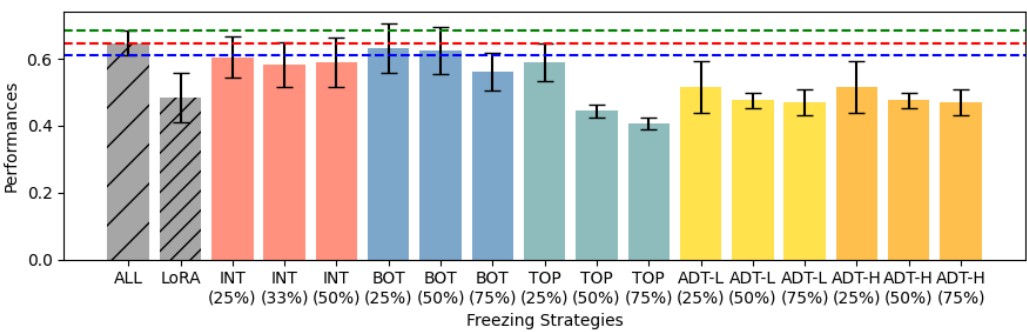

(b) F1-score performance of MiniCPM on the CB task

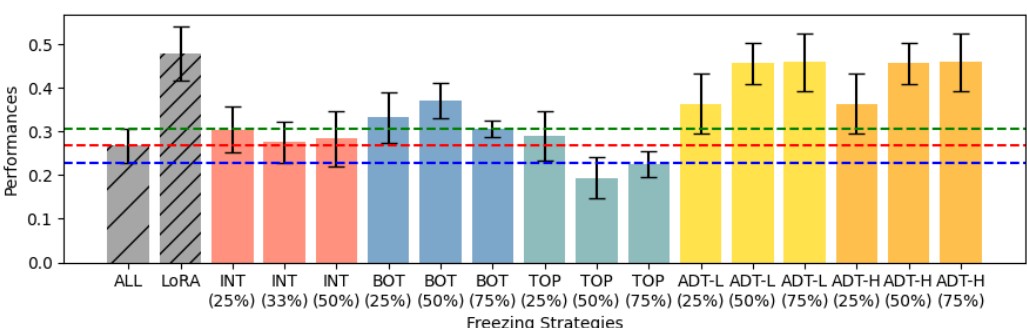

(c) Accuracy performance of MiniCPM on the WNLI task

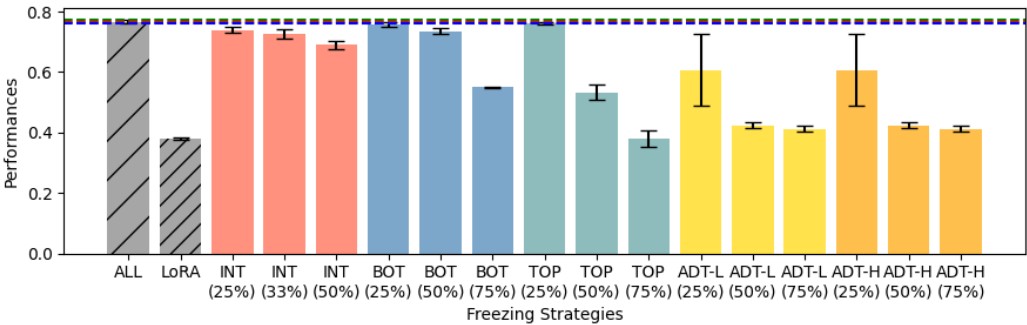

(d) Accuracy performance of MiniCPM on the MNLI task

Figure 11: Accuracy performance of MiniCPM. The same colors indicate the same strategy. The gray hatched bar represents the baseline performance achieved by fine-tuning all layers of the model. The red dashed line represents the average performance of fine-tuning across all layers, The blue dashed line represents the mean performance minus one standard deviation, while the green dashed line shows the mean performance plus one standard deviation when all layers are fine-tuned.

