# OpenReview forum: "Exploring Selective Layer Freezing Strategies in Transformer Fine-Tuning: NLI Classifiers with Sub-3B Parameter Models"
_ICLR.cc/2025/Conference — ICLR 2025 Conference Withdrawn Submission_

### Official Review · Reviewer_LLYf · 2024-10-30

**Soundness:** 1
**Presentation:** 3
**Contribution:** 2
**Rating:** 3
**Confidence:** 4

**Summary:**

The proposed parameter-efficient fine-tuning technique freezes entire layers. The authors compare different layer selection strategies (top, bottom, interval, adaptive). The authors evaluate across a subset of [super]glue and 3 different models.

**Strengths:**

- Simple PET strategy
- Results include standard deviations
- Some setups seem to improve over full fine-tuning

**Weaknesses:**

- IIUC each experiment used 100 steps and a learning rate of 5e-5, i.e. Fig. 3 does not show accuracies after convergence on the dev set? Training until convergence and tuning the learning rate for each setup would make the evaluation much more convincing.
- The dataset choice is very odd. Why not use the full GLUE or SuperGLUE benchmarks? And why only using 3% of QNLI and MNLI? This makes the results hard to compare with related work.
- Similar approaches like BitFit of Ben Zaken et al. or gradual freezing of Tang et al. are cited, but not compared with. I think that BitFit (freezing everything except bias terms) is misleadingly described as having a high "complexity of freezing techniques" (L049, L136). IMO BitFit is easy enough to implement, and should be used as a baseline here.
- I don't really see any benefit of the ADT strategy. It is slower than full fine-tuning (Fig. 4), is worse in terms of performance, and I can imagine that it is using more memory than full fine-tuning, although unfortunately Fig. 5 does not include ADT

**Questions:**

- What is the memory footprint of ADT?

---

### Official Review · Reviewer_GM6U · 2024-10-31

**Soundness:** 1
**Presentation:** 2
**Contribution:** 3
**Rating:** 6
**Confidence:** 4

**Summary:**

The authors present a straightforward yet effective method, Layer Freezing, to fine-tune large language models (LLMs) by selectively freezing and training only certain transformer layers. Their study demonstrates that freezing the bottom or top 25-50% of layers improves parameter efficiency, reduces GPU memory usage, speeds up training, and enhances task performance compared to full fine-tuning and LoRA.

The methodology involves three primary freezing strategies:

Bottom-Up Freezing: Freezes lower layers to retain linguistic expressiveness and adapts higher layers for task-specific needs.
Top-Down Freezing: Freezes upper layers to hold high-level concepts and fine-tunes the lower layers.
Interval Freezing: Freezes every nth layer to balance updates across the model.
To validate the approach, the authors conduct experiments across various NLI tasks, demonstrating Layer Freezing’s superiority over LoRA and other baselines in terms of memory efficiency and fine-tuning speed.

**Strengths:**

The paper introduces an efficient fine-tuning approach that is clear, practical, and easy to implement without complex modifications.
Extensive experiments across multiple NLP tasks and models show that Layer Freezing can achieve comparable or better performance than existing methods like LoRA while significantly reducing memory and time costs.
The study’s methodology could extend easily to other LLM tasks, enhancing its potential impact on practical NLP applications.

**Weaknesses:**

Limited Novelty in Concept: The approach of selectively freezing layers has been explored in prior studies. Although the method is well-executed here, the conceptual innovation is relatively incremental compared to related work in parameter-efficient tuning methods.
Restrained Task Scope: The authors focused primarily on NLI tasks. Evaluating the freezing techniques on a broader range of NLP tasks, such as text generation or named entity recognition, would better support the generalizability of the approach.
Baseline Comparisons: The study compares the proposed strategies primarily with LoRA and full fine-tuning. Including comparisons with other recent parameter-efficient fine-tuning techniques, like BitFit or parameter pruning methods, would provide a more comprehensive evaluation.
Adaptive Freezing Strategy Effectiveness: The Adapted Freezing strategy, which dynamically selects layers based on weight change, does not demonstrate a clear advantage over static methods. Further analysis or refinement could improve its applicability.

**Questions:**

Can you provide additional empirical results on the memory and computational savings specific to each freezing strategy? This would help clarify which techniques are most effective in different resource-limited settings.
How well does your approach generalize to other types of tasks outside of NLI? Results on tasks like summarization or machine translation would strengthen the case for the broader applicability of selective layer freezing.
For the Adaptive Freezing strategy, how does the layer selection process impact overall training time, particularly for larger models?
Have you explored freezing layers in non-sequential patterns, such as random selection, and if so, how does that compare to Interval Freezing in terms of efficiency?

---

### Official Review · Reviewer_GLr9 · 2024-11-04

**Soundness:** 2
**Presentation:** 3
**Contribution:** 2
**Rating:** 3
**Confidence:** 3

**Summary:**

This paper investigates optimal layer-wise fine-tuning strategies for 3B language models applied to NLI classification tasks. The selective fine-tuning approach involves freezing specific layers while optimizing the remaining ones, offering a parameter-efficient alternative to full-parameter fine-tuning. The study focuses on two main methods: fixed freezing and adaptive freezing. Results from NLI benchmarks indicate that freezing the bottom 25% or 50% of layers yields the best performance-efficiency trade-off, supporting that this strategy as an ideal practical solution in the field of parameter-efficient fine-tuning.

**Strengths:**

1. This paper provides valuable practical insights for researchers and practitioners looking to fine-tune 3B models on NLI downstream tasks within resource-constrained environments.
2. The writing is clear and well-structured, making it easy for readers to follow the exploration scope and findings.

**Weaknesses:**

1. The selection of selective fine-tuning baselines could be more comprehensive. Many selective fine-tuning methods based on layer freezing exist [1-3], yet the authors do not discuss these approaches. At a minimum, a cited work [4] in the related work section should be included in the experiments.
2. Although the training memory usage is reported, the number of trainable parameters is not provided. This omission could undermine the fairness of comparisons. For instance, it would be beneficial to adjust the training settings (rank, alpha) of LoRA to ensure comparable baselines with the selective layer freezing methods.
3. In Section 4.1, the reason for using randomly selected data in the experiments is unclear.
4. The scale of data used for fine-tuning is limited. A discussion on how data scaling affects the freezing strategies would enhance the depth of the experiments.
5. Typos:
- Line 233: Gemma-2b(Gemma) -> Gemma-2b (Gemma)
- Line 233: MiniCPM-2b-128k(MiniCPM) -> MiniCPM-2b-128k (MiniCPM)
- Line 239: SuperLUEWang et al. (2019) -> SuperLUE (Wang et al., 2019)

References

- [1] Liu, Y., Agarwal, S., & Venkataraman, S. (2021). Autofreeze: Automatically freezing model blocks to accelerate fine-tuning. arXiv preprint arXiv:2102.01386.
- [2] Vrbančič, G., & Podgorelec, V. (2020). Transfer learning with adaptive fine-tuning. IEEE Access, 8, 196197-196211.
- [3] Zhu, L., Hu, L., Lin, J., & Han, S (2023). LIFT: Efficient Layer-wise Fine-tuning for Large Model Models.
- [4] Tang, H., Chen, J., Zhang, W., & Guo, Z. (2024). Training Acceleration Method Based on Parameter Freezing. Electronics, 13(11), 2140.

**Questions:**

Q1: Would freezing the bottom 25% or 50% of layers also be suitable choices for models larger than 3B?

Q2: In Section 4.3, will the models’ coverage be determined by training them for a total of 100 steps across all experiments?

---

### Official Review · Reviewer_eE5U · 2024-11-06

**Soundness:** 2
**Presentation:** 3
**Contribution:** 1
**Rating:** 3
**Confidence:** 4

**Summary:**

This paper proposes to reduce memory consumption during LLM training. The proposed method tracks weight changes and freezes the top N layers with either the largest weight change (Adapted High; ADT-H) or the smallest weight change (Adapted Low; ADT -L). The experiments were conducted on relatively small LLMs (model size less than 3B), such as Gemma-2b, Phi-2, and MiniCPM-2b-128k.

**Strengths:**

1. The paper is well structured and easy to follow.
2. Figures are well produced for quick understanding.

**Weaknesses:**

1. The proposed method is crude. Though this paper claims to reduce memory usage during training, the layers to be freeze are simply decided by the L2 norm over the weight values. If this design is an optimal choice, the authors should provide experiments to compare the proposed method with the other layer freezing approaches.
2. The proposed method is not effective. Baselines (such as BOT 25% or TOP 25%) outperform the proposed method in Figure 3.
3. Limited tasks tested. The authors only focus on experiments on NLI tasks. However, experiments on more evaluation datasets should be included.

**Questions:**

1. Is there a reason for using only NLI tasks in the experiments?
2. Why does the proposed method not appear in Figure 5?
3. Considering the computational graphs during back-propagation, freezing top layers should have lower memory usage compared with freezing bottom layers. However, higher memory usage of freezing top layers (TOP) is observed in comparison of freezing bottom layers (BOT). Can you provide an explanation for this circumstance?
4. According to L176, weight change is tracked for **only 5 training steps**. What will happen if 5 steps are not enough to observe the weight difference?

---

### Comment · Area_Chair_ghGz · 2024-11-21
**Reminder: Please respond and update the score if necessary**

Dear Reviewers,

Kindly ensure that you respond proactively to the authors' replies (once they are available) so we can foster a productive discussion. If necessary, please update your score accordingly. We greatly appreciate the time and effort you’ve dedicated to the review process, and your contributions are key to making this process run smoothly.

Thank you,

AC

---

### Note · Authors · 2025-08-29

I have read and agree with the venue's withdrawal policy on behalf of myself and my co-authors.

---

### Meta-Review · Area_Chair_ghGz · 2024-12-19

**Metareview:**

The paper presents a method to reduce memory consumption during training of large language models by selectively freezing specific transformer layers, tracking weight changes for optimization. This approach, tested on models like Gemma-2b and MiniCPM-2b, focuses on improving parameter efficiency for around 3B parameter models, particularly in natural language inference tasks. Three primary freezing strategies are introduced: Bottom-Up, Top-Down, and Interval Freezing. Experiments show that selectively freezing 25-50% of layers offers the best performance-efficiency trade-off, outperforming methods like LoRA in memory efficiency and fine-tuning speed. This strategy significantly reduces GPU memory usage and accelerates training while enhancing task performance.

Reviewers have raised concerns about the non-standard dataset split, and noted that freezing parameters is a technique already well-explored in the literature, indicating limited novelty from a technical standpoint. Consequently, I recommend rejecting the paper, as it does not meet ICLR's standards.

**Additional Comments On Reviewer Discussion:**

There have been no observed changes or developments, as both the reviewers and the authors have not engaged in any follow-up discussions. I have encouraged them to initiate a dialogue during the author response period.

---

### Decision · Program_Chairs · 2025-01-22

Reject